# Inhibitory Effects of Shikonin Dispersion, an Extract of *Lithospermum erythrorhizon* Encapsulated in β-1,3-1,6 Glucan, on *Streptococcus mutans* and Non-Mutans Streptococci

**DOI:** 10.3390/ijms25021075

**Published:** 2024-01-16

**Authors:** Ryota Nomura, Yuto Suehiro, Fumikazu Tojo, Saaya Matayoshi, Rena Okawa, Masakazu Hamada, Shuhei Naka, Michiyo Matsumoto-Nakano, Rika Unesaki, Kazuya Koumoto, Keiko Kawauchi, Takahito Nishikata, Tatsuya Akitomo, Chieko Mitsuhata, Masatoshi Yagi, Toshiro Mizoguchi, Koki Fujikawa, Taizo Taniguchi, Kazuhiko Nakano

**Affiliations:** 1Department of Pediatric Dentistry, Osaka University Graduate School of Dentistry, Suita 565-0871, Osaka, Japan; suehiro.yuto.dent@osaka-u.ac.jp (Y.S.); u181758a@ecs.osaka-u.ac.jp (F.T.); matayoshi.saaya.dent@osaka-u.ac.jp (S.M.); okawa.rena.dent@osaka-u.ac.jp (R.O.); nakano.kazuhiko.dent@osaka-u.ac.jp (K.N.); 2Joint Research Laboratory of Next-Generation Science for Oral Infection Control, Osaka University Graduate School of Dentistry, Suita 565-0871, Osaka, Japan; yagi@pharmacrea.com (M.Y.); t_mizoguchi@takeda-co.com (T.M.); k_fujikawa@takeda-co.com (K.F.); taniguchi@pharmacrea.com (T.T.); 3Department of Pediatric Dentistry, Graduate School of Biomedical and Health Sciences, Hiroshima University, Hiroshima 734-8553, Hiroshima, Japan; takitomo@hiroshima-u.ac.jp (T.A.); chiekom@hiroshima-u.ac.jp (C.M.); 4Department of Oral & Maxillofacial Oncology and Surgery, Osaka University Graduate School of Dentistry, Suita 565-0871, Osaka, Japan; hamada.masakazu.dent@osaka-u.ac.jp; 5Department of Pediatric Dentistry, Okayama University Graduate School of Medicine, Dentistry and Pharmaceutical Sciences, Okayama 700-8558, Okayama, Japan; nshuhei@okayama-u.ac.jp (S.N.); mnakano@cc.okayama-u.ac.jp (M.M.-N.); 6Faculty of Frontiers of Innovative Research in Science and Technology (FIRST), Konan University, Kobe 650-0047, Hyogo, Japan; unerika15@gmail.com (R.U.); koumoto@konan-u.ac.jp (K.K.); kawauchi@konan-u.ac.jp (K.K.); nisikata@konan-u.ac.jp (T.N.); 7Pharmacrea Kobe Co., Ltd., Kobe 651-0085, Hyogo, Japan; 8TSET Co., Ltd., Kariya 448-0022, Aichi, Japan

**Keywords:** shikonin, dispersion, *Streptococcus mutans*, non-mutans streptococci

## Abstract

Shikonin is extracted from the roots of *Lithospermum erythrorhizon*, and shikonin extracts have been shown to have inhibitory effects on several bacteria. However, shikonin extracts are difficult to formulate because of their poor water solubility. In the present study, we prepared a shikonin dispersion, which was solubilized by the inclusion of β-1,3-1,6 glucan, and analysed the inhibitory effects of this dispersion on *Streptococcus mutans* and non-mutans streptococci. The shikonin dispersion showed pronounced anti-*S. mutans* activity, and inhibited growth of and biofilm formation by this bacterium. The shikonin dispersion also showed antimicrobial and antiproliferative effects against non-mutans streptococci. In addition, a clinical trial was conducted in which 20 subjects were asked to brush their teeth for 1 week using either shikonin dispersion-containing or non-containing toothpaste, respectively. The shikonin-containing toothpaste decreased the number of *S. mutans* in the oral cavity, while no such effect was observed after the use of the shikonin-free toothpaste. These results suggest that shikonin dispersion has an inhibitory effect on *S. mutans* and non-mutans streptococci, and toothpaste containing shikonin dispersion may be effective in preventing dental caries.

## 1. Introduction

More than 700 bacterial species, 100 fungal species, and hundreds of viral genotypes reside in the human oral cavity [1]. Streptococci are the predominant bacteria in large areas of the oral cavity, including teeth, gingiva, saliva, and tongue, accounting for approximately 15% to 60% of all bacteria [2]. When biofilm homeostasis is disrupted and a severe shift occurs, acid-adapted bacterial species including non-mutans streptococci are thought to emerge from the oral commensal bacteria [3,4,5]. Such acid-adapted bacteria acquire a selective advantage over other species, resulting in dental caries lesions [3,4,5].

When an acidic environment is formed within the biofilm, *Streptococcus mutans* also increases, leading to severe dental caries with loss of minerals on the tooth surface [4]. Based on differences in the structure of serotype-specific polysaccharides on the cell surface, *S. mutans* is serologically classified into four groups: *c*, *e*, *f*, and *k* [6]. The distribution of these serotypes among all known strains is approximately 80% type *c* and approximately 20% type *e*, while the distribution of types *f* and *k* is <5% [6]. Serotypes *f* and *k* strains show mutations in major surface proteins and have a collagen-binding property, associated with the development of cardiovascular diseases such as infective endocarditis [7].

Many antimicrobial agents are effective in preventing dental caries, but there are occasional problems such as the emergence of resistant bacteria and drug allergies [8]. The dental caries-inhibiting effects of plant-derived ingredients are attracting attention in terms of their ability to overcome these problems [9]. Shikonin is a purple dye extracted from the roots of *Lithospermum erythrorhizon* [10], a perennial plant belonging to the family Boraginaceae. Shikonin has been shown to have various beneficial biological activities, such as wound healing, antitumor effects, and improvement of allergic dermatitis [10,11]. Shikonin also has an inhibitory effect on bacteria that cause human infections, which is more pronounced for Gram-positive bacteria than for Gram-negative bacteria [12]. However, the inhibitory effect of shikonin towards oral streptococci (which are Gram-positive bacteria) has not yet been clarified.

The use of shikonin has been limited by its insolubility in water (and other solvents) [10]. Therefore, technology has been developed to solubilize shikonin by encapsulating it in certain substances [10]. In the present study, shikonin was solubilized in water by the inclusion of β-1,3-1,6 glucan. We analysed the inhibitory effect of the resulting shikonin dispersion on the survival, growth, and biofilm formation of *S. mutans* strains and non-mutans streptococci. In addition, we analysed the effect of toothpaste containing shikonin on the number of *S. mutans* in the oral cavity.

## 2. Results

### 2.1. Preparation and Concentration Determination of Shikonin Dispersion

The shikonin dispersion exhibits a structure in which the shikonin molecules are incorporated into the triple-chain structure of β-1,3-1,6 glucan (Figure 1A). The shikonin dispersion was prepared by mixing shikonin extract with aqueous β-1,3-1,6 glucan solution using a wet-milling method. The shikonin extract is poorly water soluble, but shikonin extract encapsulated in β-1,3-1,6 glucan is soluble in water and can be used as a shikonin dispersion. This shikonin dispersion was diluted and used for subsequent analyses. Figure 1B shows a saturated solution of shikonin extract in water and shikonin dispersion prepared by dilution of the stock solution to 0.20 mM. The absorption spectrum of the 0.20 mM shikonin dispersion showed a strong and characteristic absorption band at 490 nm (Figure 1C). The absorbance of the saturated solution of shikonin extract in water was very low (0.014) compared with that of the 0.20 mM shikonin dispersion (1.391), indicating that the water solubility of shikonin was dramatically increased by encapsulation in β-1,3-1,6 glucan. The absorption spectrum of the glucan suspension had no strong absorption band (Appendix A). The concentration of shikonin in the dispersion stock solution was quantified using a calibration curve for shikonin in 1.0 M NaOH, in which the hydroxyl groups of both shikonin and glucan dissociate and produce anionic charges leading to dispersion as monomers (Appendix A). The shikonin concentration was found to be 1.12 ± 0.09 mM and this shikonin dispersion was diluted and used for subsequent analyses.

### 2.2. Inhibitory Effects of Shikonin Dispersion on S. mutans

The inhibitory effects of shikonin dispersion (shikonin concentrations 0, 0.2, 2, 20, and 200 μM) on *S. mutans* strain MT8148 adjusted to 1.0 × 10^9^ colony-forming units (CFU)/mL PBS were investigated as a function of time. After 3 h, the viability of *S. mutans* MT8148 in the presence of 2, 20, and 200 μM shikonin dispersion was approximately 50%, 10%, and 5%, respectively (Figure 2A). The survival rate of strain MT8148 in the presence of ≥2 μM shikonin dispersion decreased to <1% after 6 h. Even a low concentration of shikonin dispersion (0.2 μM) inhibited strain MT8148 in a time-dependent manner; the survival rate was <10% after 24 h. In contrast, β-1,3-1,6 glucan and arginine suspensions did not affect the bacterial number of *S. mutans* (Appendix A).

Next, the inhibitory effect after 12 h of each concentration of shikonin dispersion on *S. mutans* of different serotypes (i.e., serotypes *c*, *e*, *f*, and *k*) was analysed. The survival rate of all *S. mutans* serotypes was <50% in the presence of 0.2 μM shikonin dispersion. There was a significant inhibitory effect compared with the absence of the shikonin dispersion (*p* < 0.01) (Figure 2B). The survival rate of *S. mutans* in the presence of ≥2 μM shikonin dispersion was <5% (*p* < 0.001).

### 2.3. Inhibitory Effects of Shikonin Dispersion on the Growth of S. mutans

The inhibitory effect of each concentration of shikonin dispersion on the growth of *S. mutans* strain MT8148, which was added to brain–heart infusion (BHI) broth at a concentration of 1.0 × 10^7^ CFU/mL, was examined as a function of time. The growth of strain MT8148 was inhibited after 6 h by 200 μM shikonin dispersion (Figure 3A). For 0.2–20 μM shikonin dispersion, only 20 μM shikonin dispersion inhibited the growth of MT8148, and this was only observed at 24 h. The growth of different serotypes of *S. mutans* was significantly inhibited by 200 μM shikonin dispersion compared with the absence of shikonin dispersion (*p* < 0.001) (Figure 3B). Only *S. mutans* serotypes *f* and *k*, which are minor serotypes in the oral cavity, showed significant growth inhibition by 20 μM shikonin dispersion (*p* < 0.001).

### 2.4. Inhibitory Effects of Shikonin Dispersion on Biofilm Formation by S. mutans

Biofilm formation by *S. mutans* strain MT8148 (1.0 × 10^7^ CFU/mL) cultured in BHI broth containing 1% sucrose was analysed in the presence of each concentration of shikonin dispersion. Confocal laser scanning microscopy images showed that 20 μM shikonin dispersion drastically decreased the biofilm formation, and almost no bacteria were observed in the presence of 200 μM shikonin dispersion (Figure 4A). The quantity of biofilm formed by each serotype of *S. mutans* was significantly inhibited in the presence of ≥20 μM shikonin dispersion compared with the absence of shikonin dispersion (*p* < 0.001) (Figure 4B). Under conditions without sucrose, the biofilm formation ability of *S. mutans* was markedly reduced, and the anti-biofilm effect of shikonin was not observed (Appendix A).

### 2.5. Inhibitory Effects of Shikonin Dispersion on Non-Mutans Streptococci

The inhibitory effects of shikonin dispersion on non-mutans streptococci, such as *Streptococcus sanguinis*, *Streptococcus oralis*, *Streptococcus gordonii*, and *Streptococcus salivarius*, were analysed. In the antimicrobial assay, after 12 h, 0.2 μM shikonin dispersion significantly inhibited only *S. gordonii* compared with the absence of shikonin dispersion (*p* < 0.01) (Figure 5A). *S. oralis* and *S. salivarius* were significantly inhibited by the 2 µM shikonin dispersion (*p* < 0.01), and *S. sanguinis* was significantly inhibited only by the 200 µM shikonin dispersion (*p* < 0.001). In bacterial growth tests, after 12 h, 20 μM shikonin dispersion significantly inhibited *S. sanguinis* and *S. salivarius* compared with the absence of shikonin dispersion (*p* < 0.001) (Figure 5B), and 200 μM shikonin dispersion significantly inhibited all the non-mutans streptococci compared with the absence of shikonin dispersion (*p* < 0.001).

### 2.6. Effect of Shikonin Dispersion-Containing Toothpaste on the Number of S. mutans in the Oral Cavity

We prepared toothpaste with or without 5 μM shikonin dispersion. Twenty subjects brushed their teeth with shikonin-containing and shikonin-free toothpaste, respectively, for 1 week, with a 1 week gap between the use of the two toothpastes (Figure 6A). Twenty subjects were divided into two groups and asked to use the two toothpastes in a different order in each group. Saliva specimens were collected before and after toothbrushing for 1 week to determine the number of *S. mutans*. The number of *S. mutans* did not change when shikonin-free toothpaste was used (Figure 6B), whereas a significant decrease in the number of *S. mutans* was observed when toothpaste containing shikonin was used (*p* < 0.05) (Figure 6C). When the subjects were divided into Group 1 and Group 2, there were no significant differences in bacterial number before and after tooth brushing (Appendix A).

## 3. Discussion

In the present study, we synthesized a soluble dispersion of shikonin by encapsulating it in β-1,3-1,6 glucan. The shikonin dispersion showed inhibitory effects on *S. mutans* and non-mutans streptococci. In addition, the number of *S. mutans* was decreased in the oral cavity of subjects who brushed their teeth with toothpaste containing shikonin dispersion, indicating that shikonin dispersion may be clinically effective in the prevention of dental caries.

Shikonin has a variety of medical benefits such as wound healing and antitumor effects [10], and it is widely applied in creams and ointments because of its high lipophilicity and low water solubility. However, the poor water solubility of shikonin makes it difficult to use as a pharmaceutical raw material or to administer orally [13]. Therefore, technology has been developed to solubilize shikonin [10]. In the present study, we used edible β-1,3-1,6 glucan, a polysaccharide found in fungi and seaweeds [14,15], as an inclusion agent for shikonin, and we successfully prepared a water-soluble shikonin dispersion.

Shikonin has an inhibitory effect on bacteria that cause systemic infections, including *Staphylococcus aureus*, *Enterococcus faecium*, and *Helicobacter pylori* [16]. In addition, shikonin can inhibit *Streptococcus* species indigenous outside the oral cavity, such as *S. pneumoniae*, which causes pneumonia, and *S. agalactiae*, which is associated with bacterial meningitis in newborns [17,18]. However, no studies have focused on the effects of shikonin on oral bacteria, including oral streptococci. Therefore, we decided to analyse the inhibitory effect of shikonin on *S. mutans* and non-mutans streptococci.

Dental caries are considered to be an endogenous infection because all the bacteria associated with dental caries belong to the normal oral microbiota [3,4,5]. Non-mutans streptococci can adapt to acid and survive in the acidification stage when homeostasis in the biofilm becomes unstable [3,4,5]. In such an environment, *S. mutans* and other acidic bacteria increase and promote dental caries’ progression due to the loss of minerals on the tooth surface. Therefore, we decided to analyze the effect of Shikonin on *S. mutans* and non-mutans streptococci, which are closely related to dental caries.

Antimicrobial substances, such as cetylpyridinium chloride and triclosan, are widely used in oral self-care products [19]; they have high antimicrobial activity and pose little risk to the human body when used at the correct dose [20,21]. However, there are some reports on the toxicity of these substances [22,23,24,25]. In contrast, in an experiment using rats, no toxicity was observed even when shikonin was orally administered at concentrations equivalent to or higher than those at which cetylpyridinium chloride or triclosan caused abnormalities [26]. Therefore, shikonin has a better safety profile than existing antimicrobial substances in toothpaste.

The anti-biofilm effect of shikonin was stronger than the growth inhibitory effect. This result was similar to the inhibitory effect of S-PRG filler, a bioactive functional glass, on *S. mutans* [27]. In addition, the low concentration of a mouth rinse formulated with chlorhexidine gluconate did not affect the ability of *S. mutans* growth but showed the anti-biofilm ability of the bacteria [28]. Although shikonin, S-PRG filler, and a mouth rinse formulated with chlorhexidine gluconate have no properties in common, they all have a stronger inhibitory effect on the biofilm-formation ability of *S. mutans* than on its growth ability. S-PRG filler caused changes in the expression of *S. mutans* genes [27], and the effect of shikonin on *S. mutans* gene expression should also be investigated in the future.

In the present study, we produced a toothpaste containing only 5 µM shikonin (150 µg per 100 g of toothpaste) and showed that this toothpaste was effective in reducing the number of *S. mutans* in the oral cavity. Although dental caries have decreased in Japan in recent years, oral conditions in many developing countries in Asia are much worse than in Japan [29,30]. Toothpaste containing shikonin may be helpful for such subjects with poor oral conditions. Because this small-scale clinical study confirmed the effect of shikonin, a larger-scale and more detailed study should be conducted in the future.

## 4. Materials and Methods

### 4.1. Preparation of Shikonin Dispersion

Shikonin was provided by Pharmacrea Kobe Co., Ltd. (Hyogo, Japan) and β-1,3-1,6 glucan was provided by Osaka Soda Co., Ltd. (Osaka, Japan). Inclusion dispersion of shikonin in β-1,3-1,6 glucan was performed using a wet-milling method by Fuji Pigment Co., Ltd. (Hyogo, Japan). Briefly, β-1,3-1,6 glucan (3.0 g) and arginine (3.0 g) were mixed with 94 mL of demineralized water and stirred using a homo disper (PRIMIX Corporation, Hyogo, Japan) at 100 g for 3 h at 85 °C to obtain a 3.0% (*w*/*v*) of β-1,3-1,6 glucan solution. The glucan solution (0.52 g) was diluted with 9.47 mL deionized water and mixed with 0.015 g shikonin extract (powder). The solution was dispersed using a paint conditioner (Red Devil Inc., Pryor, OK, USA) with zirconia beads as the medium for 2 h to give a shikonin dispersion.

### 4.2. Bacterial Strains and Growth Conditions

Table 1 lists the bacterial strains used in this study. A total of four *S. mutans* strains of different serotypes (*c*, *e*, *f*, and *k*) were selected from among our laboratory stocks [31,32]. The non-mutans streptococci used in this study were *S*. *sanguinis* (ATCC 10556), *S*. *oralis* (ATCC 10557), *S*. *gordonii* (ATCC 10558), and *S*. *salivarius* (HHT), which were selected from among our laboratory stocks [33]. *S. mutans* strains were cultured on Mitis Salivarius (MS)-agar (Difco Laboratories, Detroit, MI, USA) plates containing bacitracin (0.2 U/mL; Sigma-Aldrich Co., St. Louis, MO, USA) and MS agar supplemented with 15% (*w*/*v*) sucrose (MSB agar). The non-mutans streptococci were cultured on MS agar plates. Colonies of each strain were inoculated into BHI broth (Difco Laboratories, Franklin Lakes, NJ, USA) and cultured under anaerobic conditions with CO_2_ at 37 °C for 18 h for use in subsequent studies.

### 4.3. Antimicrobial Activity

The antimicrobial assay was performed in accordance with a previously described method, with some modifications [34]. Briefly, cultured bacteria were collected by centrifugation at 1000× *g* at 4 °C for 10 min. The cultures were washed and resuspended in phosphate-buffered saline (PBS) to an OD_550_ value of 1.0, which corresponds to 1 × 10^9^ CFU/mL. The shikonin dispersions were then added to the bacterial suspensions at final concentrations of 0, 0.2, 2, 20, and 200 μM shikonin. Also, β-1,3-1,6 glucan and arginine, which were adjusted to the same concentration (*w*/*v*) as shikonin, were added to the bacterial suspension instead of shikonin, respectively. Bacterial suspensions were incubated at 37 °C for 1.5, 3, 6, 12, and 24 h. These bacterial suspensions were spread on MSB agar or MS agar plates, cultured anaerobically at 37 °C for 48 h, and the number of colonies was counted. Bacterial viability was calculated by counting the number of colonies as follows: (Number of colonies with shikonin dispersion)/(Number of colonies without shikonin dispersion) × 100%. All assays were carried out three times, and mean and standard deviation values were determined.

### 4.4. Bacterial Growth Assay

Growth inhibition assays were performed in accordance with a previously described method, with some modifications [35]. Briefly, cultured bacteria were added to BHI broth at a final concentration of 1.0 × 10^7^ CFU/mL with or without shikonin dispersion (0, 0.2, 2, 20, or 200 μM shikonin). Bacterial suspensions were cultured at 37 °C for 1.5, 3, 6, 12, and 24 h. These bacterial suspensions were spread on MSB agar or MS agar plates. The plates were incubated anaerobically at 37 °C for 48 h and the number of colonies was counted. All assays were carried out three times, and mean and standard deviation values were determined.

### 4.5. Microscopic Observation of In Vitro Biofilms

Structural analysis of biofilms by confocal laser scanning microscopy was performed as previously described, with some modifications [36,37]. Briefly, cultured bacteria (final concentration 1.0 × 10^7^ CFU/mL) were added to BHI broth containing 1% (*w*/*v*) sucrose, with or without shikonin dispersion (0, 0.2, 2, 20, or 200 μM shikonin). Then, 200 µL of the bacterial suspension was added to a chambered coverglass system (CultureWell™, Grace Bio Labs, Bend, OR, USA) and incubated at 37 °C for 18 h in the dark. The bacterial cells were then stained with 5 µL of 10 mM hexidium iodide (Invitrogen, Carlsbad, CA, USA) in 1 mL of Hanks’ balanced salt solution (Lonza, Walkersville, MD, USA) for 15 min at room temperature in the dark, and the chambered coverglass systems were washed with PBS. Biofilms were observed by confocal scanning laser microscopy using an LSM510 (Carl Zeiss, Oberkochem, Germany) (https://www.zeiss.co.jp/corporate/home.html, accessed on 1 November 2022) with a reflective laser beam at 543 nm.

### 4.6. Biofilm Assay

The amounts of formed biofilms were assessed by a previously described method, with some modifications [38]. Cultured bacteria (final concentration 1.0 × 10^7^ CFU/mL) were added to BHI broth containing 1% (*w*/*v*) sucrose with or without the addition of shikonin dispersion (0, 0.2, 2, 20, or 200 μM shikonin). Similarly, bacteria were cultured in BHI broth without sucrose in the presence of various concentrations of shikonin. Next, 200 µL aliquots of the bacterial suspension were added to 96-well polystyrene microtiter plates. The plates were incubated at 37 °C for 24 h and then washed three times with PBS to remove loosely bound bacteria. Biofilms were fixed with 25% (*w*/*v*) formaldehyde for 10 min and stained with 1% (*w*/*v*) crystal violet (Sigma-Aldrich) dissolved in sterilized water for 5 min at room temperature. After washing the plate three times with PBS, the biofilm was dissolved in 95% (*v*/*v*) ethanol and OD_595_ values were determined using a microplate reader (Thermo Fisher Scientific, Waltham, MA, USA). All assays were carried out five times, and mean and standard deviation values were determined.

### 4.7. Design for Human Study

This study was conducted in full adherence to the Declaration of Helsinki. The study protocol was approved by the Ethics Committee of Osaka University Graduate School of Dentistry (Approval Number: R2-E37). Prior to sample collection, all subjects were informed of the study protocol, and provided written informed consent.

This study was conducted from 12 April 2021 to 10 August 2021. In total, 40 subjects (25 men, 15 women; 18–30 years old, mean 22.1 years) were initially enrolled in the study (Figure 6A). Saliva collected from the 40 subjects was streaked on MSB agar, then bacteria were anaerobically cultured at 37 °C for 48 h, and 20 subjects (15 men, 5 women; 18–30 years old, mean 22.6 years) with saliva containing > 4.0 × 10^3^ CFU/mL *S. mutans* were selected for the clinical study. 

The 20 subjects were asked to brush their teeth twice a day, morning and night, for 1 week with toothpaste either with or without shikonin dispersion. The toothpaste was provided by TSET Co., Ltd. (Aichi, Japan). The components of the toothpaste were glycerin and β-1,3-1,6 glucan as humectants, arginine as a pH regulator, xylitol as a sweetener, sodium alginate as a binding material, and menthol as a flavouring agent. The shikonin-containing toothpaste was formulated with 5 µM of shikonin dispersion. There was an interval of 1 week between the use of each toothpaste. Twenty subjects were divided into two groups and asked to use the two kinds of toothpaste in different orders in each group. During the experiment, the subjects were instructed to refrain from using any toothpaste, mouthwash, or antimicrobial substances other than the toothpaste given to them. Saliva was collected before the use of each toothpaste, and also after 1 week of use of each toothpaste. Saliva specimens (1 mL) were serially diluted with sterile saline and streaked on MSB agar. After incubation at 37 °C for 48 h, the number of *S. mutans* colonies was counted. 

### 4.8. Statistical Analysis

GraphPad Prism 9 software (GraphPad Software Inc., La Jolla, CA, USA) was used for statistical analyses. Comparisons between groups in the effects of shikonin dispersion on survival, bacterial growth, and biofilm formation were analyzed using the Kruskal-Wallis test, followed by the Dunn test for multiple comparisons. The number of *S. mutans* before and after brushing with toothpaste was compared using the Wilcoxon sum rank test. Results were considered to be significantly different at *p* < 0.05.

## 5. Conclusions

In summary, shikonin dispersion, a plant-derived ingredient, demonstrated potential inhibitory effects on *S. mutans* and non-mutans streptococci in in vitro analysis. We have developed a toothpaste containing shikonin as a commercial product, and our clinical study showed that toothpaste containing shikonin could reduce the number of *S. mutans*. To strengthen our result, studies using this toothpaste on a larger scale of subjects and long-term follow-up on dental caries prevention will be necessary.

## Figures and Tables

**Figure 1 ijms-25-01075-f001:**
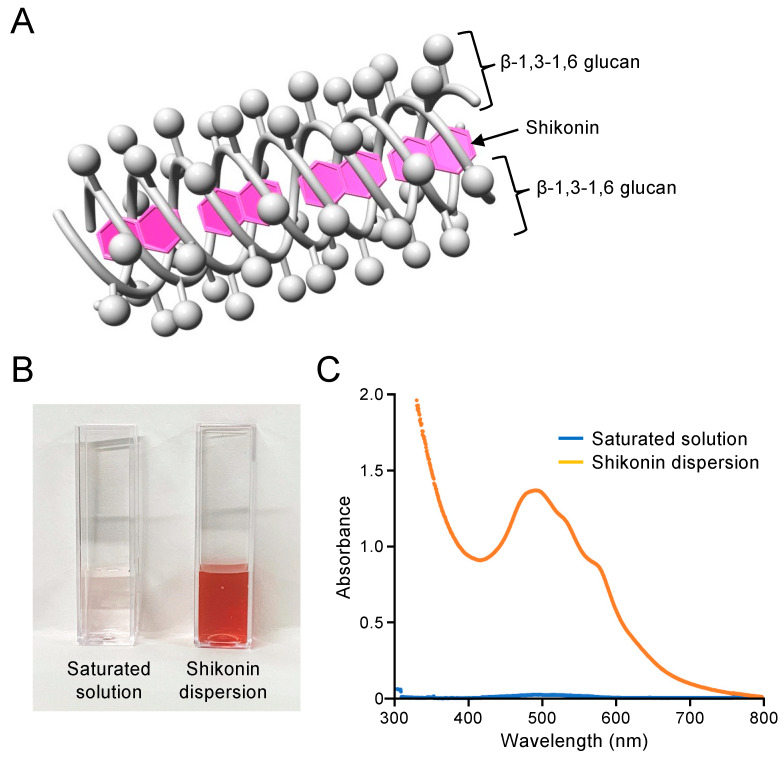
Characteristics of shikonin dispersion. (**A**) Illustration of shikonin dispersion. (**B**) A saturated solution of shikonin extract in water, and shikonin dispersion prepared by dilution of the stock solution to 0.20 mM shikonin. (**C**) Absorption spectra of the samples in panel (**B**).

**Figure 2 ijms-25-01075-f002:**
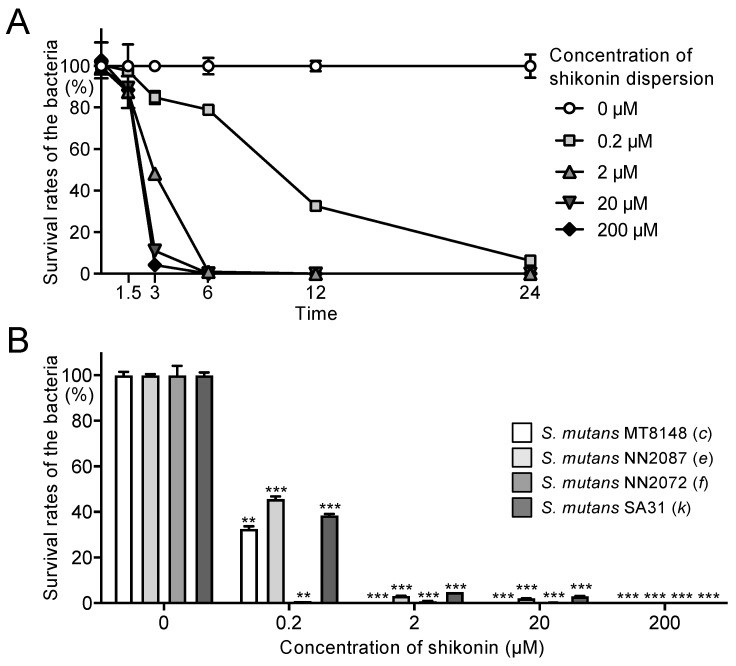
Effects of shikonin dispersion on survival of *S. mutans* strains suspended in phosphate-buffered saline (PBS). (**A**) Chronological changes in survival rate of *S. mutans* MT8148 in the presence of shikonin dispersion. (**B**) Inhibitory effect of shikonin dispersion after 12 h on the survival of *S. mutans* strains of different serotypes. Significant differences: ** *p* < 0.01 and *** *p* < 0.001 versus no shikonin dispersion.

**Figure 3 ijms-25-01075-f003:**
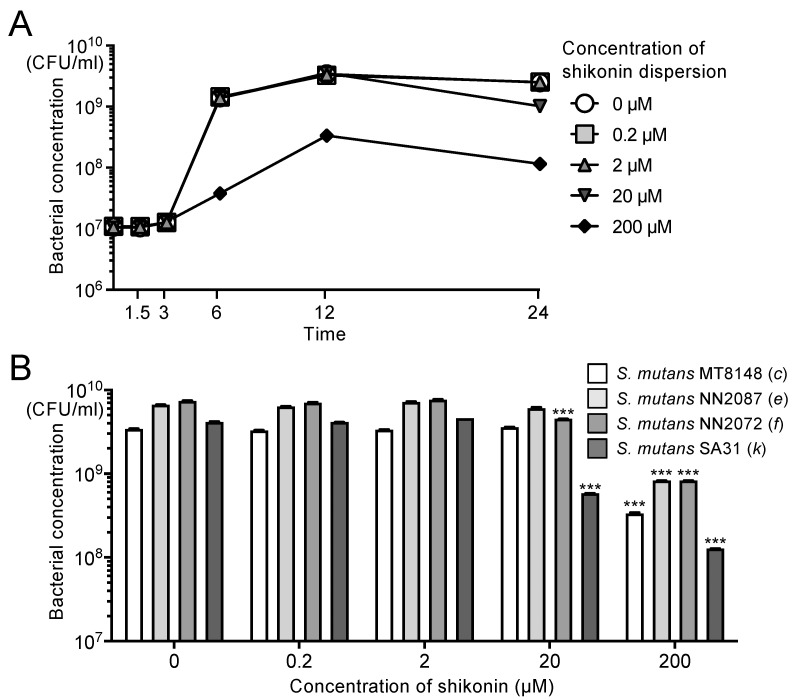
Effects of shikonin dispersion on growth of *S. mutans* strains cultured in brain–heart infusion (BHI) broth. (**A**) Chronological changes in the number [in colony-forming units (CFU)/mL] of *S. mutans* MT8148 in the presence of shikonin dispersion. (**B**) Inhibitory effect of shikonin dispersion after 12 h on the growth of *S. mutans* strains of different serotypes. Significant differences: *** *p* < 0.001 versus no shikonin dispersion.

**Figure 4 ijms-25-01075-f004:**
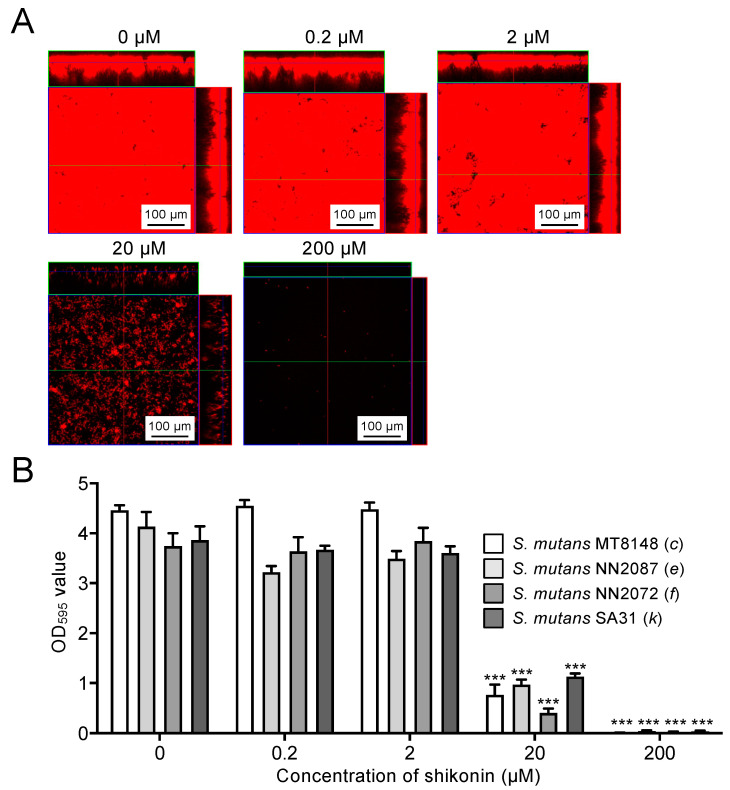
Effects of shikonin dispersion on biofilm formation by *S. mutans* strains cultured in BHI broth containing 1% sucrose. (**A**) Representative confocal scanning laser microscopy images of biofilms formed in the presence of shikonin dispersion. The red color indicates biofilm formed by *S. mutans*. The number above each image indicates the concentration of the shikonin in the dispersion. (**B**) Quantified levels of biofilm formation of *S. mutans* strains of different serotypes in the presence of shikonin dispersion. Significant differences: *** *p* < 0.001 versus no shikonin dispersion.

**Figure 5 ijms-25-01075-f005:**
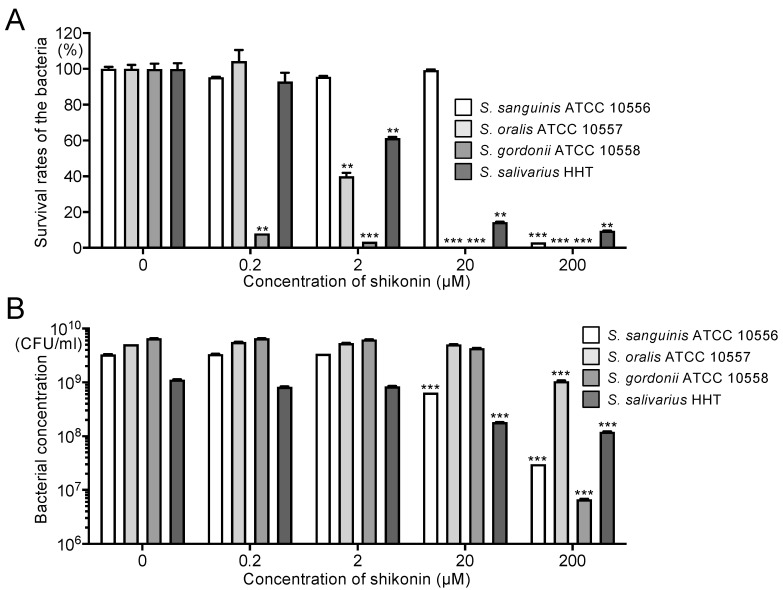
Effects of shikonin dispersion on non-mutans streptococci. (**A**) Inhibitory effects of shikonin dispersion after 12 h on non-mutans streptococci suspended in PBS. (**B**) Inhibitory effects of shikonin dispersion after 12 h on growth of non-mutans streptococci cultured in BHI broth. Significant differences: ** *p* < 0.01 and *** *p* < 0.001 versus no shikonin dispersion.

**Figure 6 ijms-25-01075-f006:**
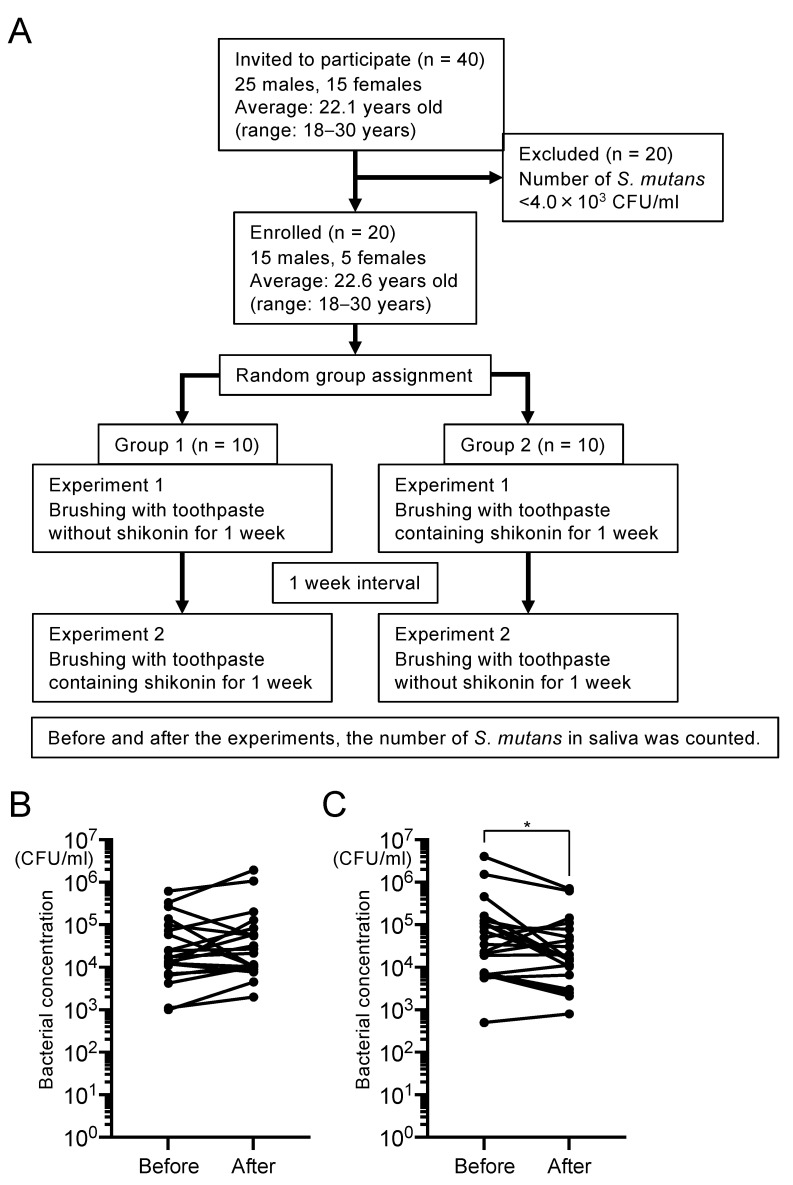
Changes in the number of *S. mutans* in saliva before and after using toothpaste with and without shikonin dispersion. (**A**) Flow diagram of the clinical study. (**B**,**C**) The number of *S. mutans* (in CFU/mL) before and after brushing with toothpaste with no shikonin dispersion for 1 week (**B**) and before and after brushing with toothpaste containing shikonin dispersion for 1 week (**C**). Significant differences: * *p* < 0.05.

**Table 1 ijms-25-01075-t001:** Bacterial strains used in the present study.

Species	Strain	Serotype	Origin	References
*S. mutans*	MT8148	*c*	Oral cavity	[31]
*S. mutans*	NN2087	*e*	Oral cavity	[28]
*S. mutans*	NN2072	*f*	Oral cavity	[28]
*S. mutans*	SA31	*k*	Oral cavity	[32]
*S. sanguinis*	ATCC 10556	-	Oral cavity	[33]
*S. oralis*	ATCC 10557	-	Oral cavity	[33]
*S. gordonii*	ATCC 10558	-	Oral cavity	[33]
*S. salivarius*	HHT	-	Oral cavity	[33]

## Data Availability

The data are available from the corresponding author upon reasonable request.

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
