# Peer review of "Inhibitory Effects of Shikonin Dispersion, an Extract of Lithospermum erythrorhizon Encapsulated in β-1,3-1,6 Glucan, on Streptococcus mutans and Non-Mutans Streptococci"

_ijms, 2024, doi:10.3390/ijms25021075_

Round 1
Reviewer 1 Report
Comments and Suggestions for Authors
The authors present an interesting manuscript in which a bioactive plant extract is prepared for use against the oral pathogen Streptococcus mutans. There are a number of points needing consideration in order to improve this manuscript for publication.
Line 85: it is not clear how the 1.12 mM value for the shikonin in the dispersion was obtained. The method used should be specified. The Methods section is helpful in describing how the dispersion was produced, but does not explain how the shikonin in the final product was quantified. If Vis spectrophotometry was used, the molar absorptivity coefficient should be mentioned and the control described below becomes very important.
Line 95: an additional control would be prefered. Although it is unlikely that the the glucan would have a significant absorbance, in Fig 1B the dispersion looks as though it may be cloudy in addition to colored, and this could contribute to an optical density. Therefore, an absorbance spectrum for a glucan control of the correct concentration is advisable.
Line 99: Appropriate controls are not included. While there is no expected effect of a mock glucan dispersion, this control is still necessary because the shikonin is not the only substance being added in this experiment (there is arginine and glucan in every treatment).
Line 139: Ideally, effects on biofilm formation can be related to bactericidal effects and growth effects to determine the specificity of the anti-biofilm effect. In this case, sucrose is added. How are the results of biofilm experiments different without the sucrose? Was the sucrose really necessary? Does BHI +sucrose+shikonin show any changes in growth inhibition as compared to Fig. 2? It appears that biofilm formation is strongly inhibited even at a shikonin concentration that has little effect on growth, a very positive finding, but more information is needed.
Fig. 3 B and Fig. 5 B: The Y axis label should be "Bacterial Concentration". The legend indicates that these are concentrations (CFU/mL). "Number" implies only CFU which would not be appropriate and is not correct according to the legend.
Line 177: Fig. 6 C is a little confusing. Since it shows the before and after S. mutans concentration following brushing with shikonin toothpaste, this includes both Group 1 Experiment 2 data and Group 2 Experiment 1 data, correct? It would be very valuable to tease out these two groups to see if the data are different. The same thing is true for Fig. 6 B; if the S. mutans is decreased in the shikonin brushers, is the one week interval between experiment sufficient? Does Group 1 Experiment 1 data look different from Group 2 Experiment 2 data? Not only are these data not shown separately, but there is no discussion of them.
Line 185: I do not have an extensive understanding of statistics, but I do not understand how a T test can be applied to these data. The range of the data is over 3 orders of magnitude, the highest CFU/mL values will have an exagerated affect on the calculations, and the standard deviations would be huge. Hopefully another reviewer can suggest a more appropriate statistical test for these data or an explanation of why this particular treatment is all right.
Line 216: This sentence is constructed in an odd way which confused me at first. The subject should be "S. mutans strains of different serotypes" were inhibited by the same concentrations in each of the tests. Rearranging the sentence puts the strains as the subject, not the tests themselves and removes some ambiguity. It is a difficult sentence to write well.
Reviewer 2 Report
Comments and Suggestions for Authors
The manuscript entitled: “Inhibitory effects of shikonin dispersion, an extract of Litho- 2
spermum erythrorhizon encapsulated in β-1,3-1,6 glucan, on Streptococcus mutans and non-mutans streptococci” is, in my opinion extremely relevant, nevertheless, in my opinion its impact can be considerably enhanced.
Introduction could be, in my opinion, considerably improved to enhance the manuscript impact. Namely, if it contained additional information regarding the S. mutans four groups. In addition, the sentence “Streptococci are the most widely distributed species in the oral cavity [1]” is in my opinion reductive. It would be extremely interesting if the authors included statistical data of the other microorganisms, and even provide geographical distribution. Furthermore, there is no information about other microorganisms besides bacteria, if they do not have any relevance in the oral cavity, this should be clearly stated and supported with adequate literature. The authors stated: “and other solvent2, such as?
Figure 1 C must include the spectra of the f β-1,3-1,6 glucan.
Species names must be stated completely in their first appearance in the abstract and in the text. After their first appearance, the genera must be abbreviated. Please ensure consistency throughout the manuscript.
The Conclusion section should be used to state the possibility of implementing the developed formulation commercially. In other words, how far is this formulation form its implementation in a commercial product? What are the required tests for its approval? Is the scale up viable? What are the most important challenges expected?
Please improve the clarity of the caption of Figure 6.
Statistical analysis, the authors should provide more information, namely regarding the tests performed to ensure the prerequisites required to perform the parametric analysis. In addition, the order of the tests should be the same as the performed tests in the manuscript.
The authors should clearly state to which corresponds the percentages: (w/w), (v/v) or (w/v), when applicable.
Line 249, please separate the unit degrees Celsius from its numerical value. Please revise throughout the manuscript.
Line 250, the authors mention: “ion exchange water” is this deionized water?
Line 263, please separate the units from the numerical value and underscore “2” from CO2.
Round 2
Reviewer 1 Report
Comments and Suggestions for Authors
looks good.
Reviewer 2 Report
Comments and Suggestions for Authors
Thank you for the performed improvements.
I would just like to recommend an economical overview in the next step of the development.